# THE DUAL INFORMATION BOTTLENECK

## ABSTRACT

The Information Bottleneck (IB) framework is a general characterization of optimal representations in large scale learning that suggests an intriguing interpretation of efficient representations in deep learning in particular. It is based on the optimal trade off between the representation complexity and accuracy, both of which are quantified by mutual information. The problem is solved by alternating projections between the *encoder* and *decoder* of the representation, which can be performed locally at each representation level. The non-parametric IB framework, however, has several drawbacks: (i) the difficult estimation of information measures in high dimensions; (ii) the inability to extend the encoder beyond the training data - the need to to know the full joint distribution of the inputs and labels; (iii) the IB does not necessarily optimize the actual prediction of unseen labels, which can be critical when training the encoder from finite samples. Here we present a new framework, the Dual Information Bottleneck (dualIB), which improves all the above drawbacks. By switching the order in the KL-divergence between the representation decoder and data, formally the dual distortion, the optimal decoder becomes the geometric rather than the arithmetic mean of the input points. This conversion preserves exponential forms of the original distribution and consequently conserves the original features of the data. We show how to solve this new representation learning formulation and illustrate it's favorable properties over the original IB when trained from a small sample. We also analyze the critical points of the dualIB and discuss their importance for the quality of this approach.

## 1    INTRODUCTION

### 1.1    THE INFORMATION BOTTLENECK METHOD

The Information Bottleneck (IB) method (Tishby et al., 1999), is an information-theoretic framework for describing efficient representations of a large scale "input" random variable X (input patterns), for predicting an "output" variable Y (desired label). In this setting the joint distribution of X and Y, $p(\mathrm{x}, \mathrm{y})$ defines the problem, or rule, and the training data are a finite sample from this distribution. In general, we assume that $p(\mathrm{y} \mid \mathrm{x})$ is strictly stochastic, and hence bounded away from $\{0, 1\}$[1]. The representation variable $\hat{\mathrm{X}}$ is in general a stochastic function of X which forms a Markov chain $\mathrm{Y} \rightarrow \mathrm{X} \rightarrow \hat{\mathrm{X}}$, and only depends on Y through the input X. We call the map $p(\hat{\mathrm{x}} \mid \mathrm{x})$ the *encoder* of the representation and denote by $p(\mathrm{y} \mid \hat{\mathrm{x}})$ the *Bayes optimal decoder* for this representation; i.e., the best possible prediction of the *desired label* Y from the representation $\hat{\mathrm{X}}$.

The IB has direct successful applications for representation learning in various domains, from vision and speech processing (Ma et al., 2019), to neuroscience (Schneidman et al., 2001), and Natural Language Processing (Li & Eisner, 2019). Due to the notorious difficulty in estimating mutual information in high dimension, variational approximations to the IB have been suggested and applied also to Deep Learning (Alemi et al., 2016; Parbhoo et al., 2018).

For large scale learning, when almost all input patterns can be considered *typical* in the information theoretic sense (Cover & Thomas, 2006), the cardinality of the typical inputs is characterized by the pattern's entropy, $H(\mathrm{X}) = -\sum_{\mathrm{x}} p(\mathrm{x}) \log p(\mathrm{x})$. Under similar typicality assumption, the mutual

---

[1]This may seem like a limitation of the method, but it can be shown that in the deterministic limit the IB is well defined. This stochastic assumption simplifies the analysis but does not pose any real limitation.

information (MI) of the encoder, $I(X; \hat{X}) = \sum_{x,\hat{x}} p(x, \hat{x}) \log \frac{p(x|\hat{x})}{p(x)}$ characterizes the cardinality of the typical representations, $\hat{X}$. On the other hand, the mutual information of the Bayes optimal decoder, $I(Y; \hat{X}) = \sum_{y,\hat{x}} p(y, \hat{x}) \log \frac{p(y|\hat{x})}{p(y)}$, is equivalent to the cross-entropy generalization error.

The IB trade off between the encoder and decoder mutual information values is defined by the minimization of the Lagrangian:

$$\mathcal{F}[p_\beta(\hat{x} \mid x); p_\beta(\hat{x}); p_\beta(y \mid \hat{x})] = I(X; \hat{X}) - \beta I(Y; \hat{X}) , \tag{1}$$

independently over the convex sets of the normalized distributions, $\{p_\beta(\hat{x} \mid x)\}$, $\{p_\beta(\hat{x})\}$ and $\{p_\beta(y \mid \hat{x})\}$, given a positive Lagrange multiplier $\beta$. As shown in (Tishby et al., 1999; Shamir et al., 2010), this is a natural generalization of the classical concept of *Minimal Sufficient Statistics* (Cover & Thomas, 2006), where the estimated parameter is replaced by the output variable Y and *exact* statistical sufficiency is characterized by the mutual information equality: $I(\hat{X}; Y) = I(X; Y)$. The minimality of the statistics is captured by the minimization of $I(X; \hat{X})$, due to the Data Processing Inequality (DPI). However, non-trivial minimal sufficient statistics only exist for very special parametric distributions known as exponential families (Brown, 1986). Thus in general, the IB relaxes the minimal sufficiency problem to a continuous family of representations $\hat{X}$ which are characterized by the trade off between compression, $I(X; \hat{X}) \equiv I_X$, and accuracy, $I(Y; \hat{X}) \equiv I_Y$, along a convex line in the *Information-Plane* ($I_Y$ vs. $I_X$). When the rule $p(x, y)$ is strictly stochastic, the convex optimal line is smooth and each point along the line is uniquely characterized by the value of $\beta$. We can then consider the optimal representations $\hat{x} = \hat{x}(\beta)$ as encoder-decoder pairs: $(p_\beta(x|\hat{x}), p_\beta(y|\hat{x}))^2$ - a point in the continuous manifold defined by the Cartesian product of these distribution simplexes. We also consider a small variation of these representations, $\delta\hat{x}$, as an infinitesimal change in this (encoder-decoder) continuous manifold (not necessarily on the optimal line(s)).

## 1.2 IB AND RATE-DISTORTION THEORY

The IB optimization trade off can be considered as a generalized rate-distortion problem (Cover & Thomas, 2006) with the distortion function between a data point, x and a representation point $\hat{x}$ taken as the KL-divergence between their predictions of the desired label y:

$$d_{\text{IB}}(x, \hat{x}) = D[p(y \mid x) || p_\beta(y|\hat{x})] = \sum_y p(y \mid x) \log \frac{p(y \mid x)}{p_\beta(y \mid \hat{x})} . \tag{2}$$

The expected distortion $\mathbb{E}_{p_\beta(x,\hat{x})}[d_{\text{IB}}(x, \hat{x})]$ for the optimal decoder is simply the label-information loss: $I(X; Y) - I(\hat{X}; Y)$, using the Markov chain condition. Thus minimizing the expected IB distortion is equivalent to maximizing $I(\hat{X}; Y)$, or minimizing equation 1. Minimizing this distortion is equivalent to minimizing the log-loss or the cross-entropy loss, as done in most deep learning applications, and it upper-bounds other loss functions such as the $\mathcal{L}_1$-loss (due to the Pinsker inequality, or (Painsky & Wornell, 2018)). The Pinsker inequality shows that both orders of the cross-entropy upper bound to the $\mathcal{L}_1$-loss, or total-variation distance, $\min\{D[q||p], D[p||q]\} \geq \frac{1}{2\log 2}||p - q||_1^2$ .

## 1.3 THE IB EQUATIONS

For discrete X and Y, a necessary condition for the IB (local) minimization is given by the three self-consistent equations for the optimal encoder-decoder pairs, known as the IB *equations*:

$$\begin{cases} (i) & p_\beta(\hat{x} \mid x) = \frac{p_\beta(\hat{x})}{Z(x;\beta)} e^{-\beta D[p(y|x)||p_\beta(y|\hat{x})]} \\ (ii) & p_\beta(\hat{x}) = \sum_x p_\beta(\hat{x} \mid x) p(x) \\ (iii) & p_\beta(y \mid \hat{x}) = \sum_x p(y \mid x) p_\beta(x \mid \hat{x}) \end{cases} , \tag{3}$$

where $Z(x; \beta)$ is the normalization function. Iterating these equations is a generalized, Blahut-Arimoto, alternating projection algorithm (Tusnady & Csiszar, 1984; Cover & Thomas, 2006) and it converges to a stationary point of the Lagrangian, equation 1 (Tishby et al., 1999). Notice that the minimizing decoder, (equation 3-$(iii)$), is precisely the *Bayes optimal decoder* for the representation $\hat{x}(\beta)$, given the Markov chain conditions.

---

[2]Here we use the *inverse encoder*, which is in the fixed dimension simplex of distributions over X.

## 1.4 CRITICAL POINTS AND CRITICAL SLOWING DOWN

One of the most interesting aspects of the IB equations is the existence of critical points along the optimal line of solutions in the *Information-Plane* (i.e. the information curve). At these points the representations change topology and cardinality (number of clusters) (Zaslavsky & Tishby, 2019; Parker et al., 2003) and they form the skeleton of the information curve and representation space. Under the strict stochastic assumption, the information-curve is a smooth function of the Lagrange multiplier $\beta$, but its derivative may not be smooth. Critical points are bifurcations of the solutions, which are values of $\beta$ for which two different solutions (representations) co-exist. To identify such points we perform a perturbation analysis of the IB equations, as in (Zaslavsky & Tishby, 2019). Taking a small perturbation of the representation, denoted for brevity by $\delta\hat{\mathrm{x}}$, the changes in the log encoder and log decoder that satisfy equation 3 for a given $\beta$ can be determined through the nonlinear eigenvalues problems[3]:

$$\left[I - \beta C_{\mathrm{xx'}}^{\mathrm{IB}}(\hat{\mathrm{x}}, \beta)\right] \frac{\partial \log p_\beta(\mathrm{x'} \mid \hat{\mathrm{x}})}{\partial \hat{\mathrm{x}}} = 0 \,, \quad \left[I - \beta C_{\mathrm{yy'}}^{\mathrm{IB}}(\hat{\mathrm{x}}, \beta)\right] \frac{\partial \log p_\beta(\mathrm{y'} \mid \hat{\mathrm{x}})}{\partial \hat{\mathrm{x}}} = 0, \quad (4)$$

with the two square matrices $C$ defined by:

$$C_{\mathrm{xx'}}^{\mathrm{IB}}(\hat{\mathrm{x}}, \beta) = \sum_{\mathrm{y}} p(\mathrm{y} \mid \mathrm{x}) \frac{p_\beta(\mathrm{x'} \mid \hat{\mathrm{x}})}{p_\beta(\mathrm{y} \mid \hat{\mathrm{x}})} p(\mathrm{y} \mid \mathrm{x'}) \,, \quad C_{\mathrm{yy'}}^{\mathrm{IB}}(\hat{\mathrm{x}}, \beta) = \sum_{\mathrm{x}} p(\mathrm{y} \mid \mathrm{x}) \frac{p_\beta(\mathrm{x} \mid \hat{\mathrm{x}})}{p_\beta(\mathrm{y} \mid \hat{\mathrm{x}})} p(\mathrm{y'} \mid \mathrm{x}). \quad (5)$$

As shown in (Zaslavsky & Tishby, 2019), these two matrices have the same eigenvalues and have non-trivial eigenvectors (i.e., different co-existing optimal representations) precisely at the critical values of $\beta$, the bifurcation points of the IB solution. At these points the cardinality of the representation $\hat{\mathrm{X}}$ (the number of "IB-clusters") changes due to splits of clusters, resulting in topological phase transitions in the encoder. These critical points form the "skeleton" of the topology of the optimal representations. Between critical points the optimal representations change continuously (with $\beta$).

The important computational consequence of critical points is known as *critical slowing down* (Tredicce et al., 2004). For binary Y, near a critical point the convergence time, $\tau_\beta$, of the iterations of equation 3 scales like: $\tau_\beta \sim 1/(1 - \beta\lambda_2)$, where $\lambda_2$ is the second eigenvalue of either $C_{\mathrm{yy'}}^{\mathrm{IB}}$ or $C_{\mathrm{xx'}}^{\mathrm{IB}}$. At criticality, $\lambda_2(\hat{\mathrm{x}}) = \beta^{-1}$ and the number of iterations diverges. This phenomenon dominates any local minimization of equation 3 which is based on alternate encoder-decoder optimization.

To illustrate the importance of the critical points and the critical slowing-down we consider a low-dimensional problem that is easy to visualize. Taking the joint distribution, $p(\mathrm{y}, \mathrm{x})$ with a binary Y and only 5 possible inputs. For a given $\beta$, the Blahut-Arimoto iterations converge to a stationary solutions of the IB *equations*, which amounts to discrete points in the interval $[0, 1]$ - the simplex of the $p(\mathrm{y})$ distributions - for each value of the Lagrange multiplier, $\beta$.

The evolution of the optimal decoder, $p_\beta(\mathrm{y} = 0 \mid \hat{\mathrm{x}})$, $\forall \hat{\mathrm{x}} \in \hat{\mathrm{X}}$, as a function of $\beta$, is a *bifurcation diagram* (Figure 1(a)). The critical points of the IB equations correspond to bifurcations in the decoders' graph. At these critical points the non-trivial eigenvalues of $C_{\mathrm{yy'}}^{\mathrm{IB}}(\beta)$ cross the $\beta^{-1}$ value (Figure 1(b)), and the number of iterations diverges (critical slowing down) (Figure 1(c)).

## 1.5 DRAWBACKS OF THE IB

The main theoretical and practical drawback of the IB is it's apparent reliance on the explicit knowledge of the joint distribution $p(\mathrm{x}, \mathrm{y})$. In the context of representation learning this is often considered a major flaw, as we are always given only a sample from the distribution. While the generalization properties of the finite sample IB representations have been studied in the past (Shamir et al., 2010), the main problem remains the extension of the *encoder*, $p(\hat{\mathrm{x}} \mid \mathrm{x})$, when trained from a small sample, to new unobserved patterns $\mathrm{x} \in \mathrm{X}$. The standard approach in representation learning is to assume (at least implicitly) the existence of features of x with their topology, or a specific parametric form for the encoder. The original IB is completely non-parametric and does not assume any topology on the input space, beyond label similarity. A related drawback of the IB is it's poor scalability with the size of the pattern-space X, which makes the reliable estimation of the encoder mutual information impractical for large learning problems.

---

[3]We ignore here the interactions between the representations, for simplicity.

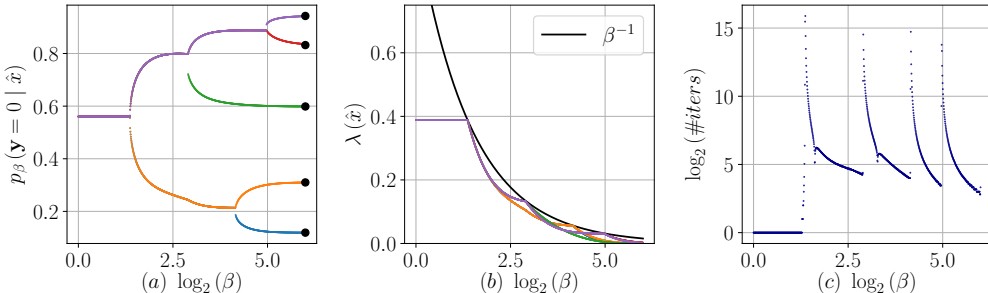

Figure 1: Example of the evolution of the IB decoder with growing $\beta$. $(a)$ The *bifurcation diagram*; each color corresponds to one component of the representation $\hat{x} \in \hat{X}$ and depicts the decoder $p_\beta(\mathbf{y} = 0 \mid \hat{x})$, in the simplex $[0, 1]$. The black dots correspond to the input distribution $p(\mathbf{y} = 0 \mid \mathbf{x})$. $(b)$ The second eigenvalue of $C_{\mathbf{yy'}}(\hat{x}; \beta)$, $\lambda_2(\hat{x})$, along with $\beta^{-1}$ as a function of $\beta$. $(c)$ Convergence time of the algorithm as a function of $\beta$.

### 1.6 CONTRIBUTIONS OF THIS WORK

Our main contribution is by providing a general formal framework resolving the encoder extension drawback of the IB. While the use of specific features and variational approximations to the IB are not new, here we take a different approach assuming that $p(\mathbf{y} \mid \mathbf{x})$ can be approximated by exponential families. This defines a new rate-distortion problem, the Dual Infromation Bottleneck (dualIB).

We show that the new dualIB formulation preserves the sufficient statistics of the original distribution for all values of the trade off parameter, $\beta$. A desired property upon predictions of a parametrized distribution. Furthermore, it obtains a significant improvement of the finite sample label prediction. We demonstrate numerically that the dualIB encoder-decoder pairs improve the prediction of unseen labels when trained from small samples, compared to the original IB. We also discuss in detail the critical points of the new formulation and their relation to those of the original IB. Interestingly, as we elaborate in the following sections, all these desirable properties are achieved by merely switching the order of the arguments in the KL-divergence of the IB distortion.

The remainder of the paper is organized as follows. We present and solve the dualIB problem in §2. We discuss the dualIB's critical points in §2.1 and prove a general relation between the critical points of the dual and original IB in §2.2. In particular, we show that the dual critical points give the best approximation to the original IB information curve. In §3 we discuss the dualIB for exponential families, dualExpIB, and demonstrate the advantages of our framework in this case. We also demonstrate numerically the improvement in the label prediction of the dualIB when trained from a small sample. We conclude in §4 with further extensions and possible applications.

## 2 THE DUALIB: MAXIMIZING THE PREDICTION INFORMATION

Supervised learning is generally separated into two phases: the training phase, where the internal representations are formed from the training data, and the prediction phase, where these representations are used to predict labels of new input patterns. We add to our Markov chain another variable, $\hat{Y}$, the *predicted label* which obtains the same values as Y, but distributed differently:

$$\overbrace{\mathbf{Y} \rightarrow \underbrace{\mathbf{X} \rightarrow \hat{\mathbf{X}}_\beta}_{\text{prediction}} \rightarrow \hat{\mathbf{Y}}}^{\text{training}}. \tag{6}$$

The left-hand part of this chain describes the representation training, and the right-hand part is the Maximum Likelihood (ML) prediction using these representations (Slonim et al., 2006). So far the prediction variable $\hat{Y}$ has not been part of the IB optimization problem. It has been implicitly assumed that the *Bayes optimal decoder*, $p_\beta(\mathbf{y} \mid \hat{x})$, which minimizes the IB distortion at a given $\beta$,

is also the best choice for making predictions of $\hat{Y}$ from the representation $\hat{X}_\beta$ through the right-hand Markov chain. That is, denoting $p_\beta(\hat{y} \mid \hat{x}) \equiv p_\beta(y \mid \hat{x})$ the ML decoder is the mixture over the internal representations:

$$p_\beta(\hat{y} \mid x) \equiv \sum_{\hat{x}} p_\beta(\hat{y} \mid \hat{x}) p_\beta(\hat{x} \mid x). \tag{7}$$

This, however, is not necessarily optimal when the encoder and decoder are trained from finite samples (Shamir et al., 2010), or when the decoder is not Bayes optimal. We define the dualIB distortion by merely switching the order of the arguments in the KL-divergence of the original IB distortion, namely:

$$d_{\text{dualIB}}(x, \hat{x}) = D[p_\beta(y \mid \hat{x}) \| p(y \mid x)] = \sum_y p_\beta(y \mid \hat{x}) \log \frac{p_\beta(y \mid \hat{x})}{p(y \mid x)} , \tag{8}$$

which in geometric terms is known as the *dual* distortion problem (Felice & Ay, 2019).

That is, given the "input" random variable X (input patterns), the "output" random variable Y (desired label) and the "representation" $\hat{X}$, the ML optimal "predicted label" $\hat{Y}$ is given by the mixture distribution, equation 7.[4]

The dualIB optimization can be written as the following rate-distortion problem:

$$\mathcal{F}^*[p_\beta(\hat{x} \mid x); p_\beta(\hat{x}); p_\beta(y \mid \hat{x})] = I(X; \hat{X}) + \beta \mathbb{E}_{p(x,\hat{x})}[d_{\text{dualIB}}(x, \hat{x})] , \tag{9}$$

with the average distortion given in terms of mutual information on $\hat{Y}$, $I(\hat{X}; \hat{Y})$ and $I(X; \hat{Y})$:

$$\mathbb{E}_{p(x,\hat{x})}[d_{\text{dualIB}}(x, \hat{x})] = \underbrace{I(\hat{X}; \hat{Y}) - I(X; \hat{Y})}_{(a)} + \underbrace{\mathbb{E}_{p(x)}[D[p_\beta(\hat{y} \mid x) \| p(\hat{y} \mid x)]]}_{(b)}$$

$$\geq I(\hat{X}; \hat{Y}) - I(X; \hat{Y}) . \tag{10}$$

This is similar to the known IB relation: $\mathbb{E}_{p(x,\hat{x})}[d_{\text{IB}}(x, \hat{x})] = I(Y; X) - I(Y; \hat{X})$ with an extra positive term $(b)$.

Both terms, $(a)$ and $(b)$, vanish precisely when $\hat{X}$ is a sufficient statistic for X with respect to $\hat{Y}$, since we can then reverse the order of X and $\hat{X}$ in the Markov chain (equation 6). This replaces the roles of Y and $\hat{Y}$ as the variable for which $\hat{X}_\beta$ are approximately minimally sufficient statistics. In that sense the dualIB shifts the emphasis from the training phase to the prediction phase. This implies that minimizing the dualIB functional maximizes the mutual information between Y and $\hat{Y}$, $I(Y; \hat{Y})$, as well as the mutual information $I(X; \hat{Y})$. This is illustrated in figure 4$(b)$-$c$.

The next theorem states the form of the solutions of *the Dual Information Bottleneck*:

**Theorem 1.** *The minima of equation 9, can be obtained by generalized Blahut-Arimoto iterations between the encoder and the decoder as in the original* IB*, with the following modifications: (i) Replace the distortion by its dual in the encoder update; (ii) Update the decoder by the encoder's "geometric" mean of the data distributions* $p(y \mid x)$.

The proof is given in §A.2.

The alternating projections between the encoder and decoder, which converge to a solution of the dualIB at a given value of the Lagrange multiplier $\beta$, are implemented by the following iterative algorithm[5]:

---

[4]We use the notation $p(\hat{y} \mid x) \equiv p(y \mid x)$, when there is no $\beta$ subscript.

[5]Unless stated otherwise, $\log = \ln \equiv \log_e$

---

**Algorithm 1** dualIB iterative algorithm

---

1: **while** $\left| p_\beta^{t+1}(\mathbf{y} \mid \hat{\mathbf{x}}) - p_\beta^t(\mathbf{y} \mid \hat{\mathbf{x}}) \right| > \epsilon$ **do**

2: $\quad Z_{\hat{\mathbf{x}}|\mathbf{x}}^{t+1}(\mathbf{x};\beta) = \sum_{\hat{x}} p_\beta^t(\hat{\mathbf{x}}) e^{-\beta D\left[ p_\beta^t(\mathbf{y}|\hat{\mathbf{x}}) \| p(\mathbf{y}|\mathbf{x}) \right]}$

3: $\quad p_\beta^{t+1}(\hat{\mathbf{x}} \mid \mathbf{x}) = \frac{p_\beta^t(\hat{\mathbf{x}})}{Z_{\hat{\mathbf{x}}|\mathbf{x}}^{t+1}(\mathbf{x};\beta)} e^{-\beta D\left[ p_\beta^t(\mathbf{y}|\hat{\mathbf{x}}) \| p(\mathbf{y}|\mathbf{x}) \right]}$

4: $\quad p_\beta^{t+1}(\hat{\mathbf{x}}) = \sum_{\mathbf{x}} p_\beta^{t+1}(\hat{\mathbf{x}} \mid \mathbf{x}) p(\mathbf{x})$

5: $\quad Z_{\mathbf{y}|\hat{\mathbf{x}}}^{t+1}(\hat{\mathbf{x}};\beta) = \sum_{\mathbf{y}} e^{\sum_{\mathbf{x}} p_\beta^{t+1}(\mathbf{x}|\hat{\mathbf{x}}) \log p(\mathbf{y}|\mathbf{x})}$

6: $\quad p_\beta^{t+1}(\mathbf{y} \mid \hat{\mathbf{x}}) = \frac{1}{Z_{\mathbf{y}|\hat{\mathbf{x}}}(\hat{\mathbf{x}};\beta)} e^{\sum_{\mathbf{x}} p_\beta^{t+1}(\mathbf{x}|\hat{\mathbf{x}}) \log p(\mathbf{y}|\mathbf{x})}$

7: **return** $p_\beta(\hat{\mathbf{x}}), p_\beta(\hat{\mathbf{x}} \mid \mathbf{x}), p_\beta(\mathbf{y} \mid \hat{\mathbf{x}})$

---

Here the $Z$'s are standard normalization factors (partition functions).
As in the IB, the encoder update (row 3) and decoder update (row 6) are the core of the algorithm.

Figure 2 illustrates the behavior of the dualIB algorithm for the same distribution as in Figure 1. The overall behavior is very similar, but the critical points (representation splits) slightly change their location. In the following sections we provide a detailed analysis and comparison between the two frameworks.

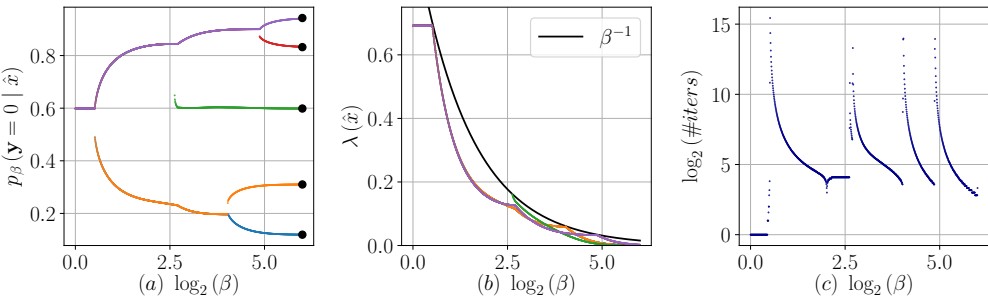

Figure 2: Same as Figure 1 for the dualIB solutions with the $C_{\mathrm{yy'}}^{\mathrm{dualIB}}$ matrix. $(a)$ The representations bifurcate at the critical values of $\beta$. $(b)$ Critical points appear when an eigenvalue (color) crosses the $\beta^{-1}$ (black) line. $(c)$ Near these points there is *critical slowing down* and the numbers of iterations of Algorithm 1 diverge.

### 2.1 THE CRITICAL POINTS OF THE dualIB

As discussed in §1.4 the skeleton of the IB optimal bound (the information curve) is constituted by the critical points in which the topology (cardinality) of the representation changes. A similar stability analysis of the dualIB equations reveals similar conditions for the critical points.

**Theorem 2.** *The* dualIB *critical points are detected by non-trivial solutions of the nonlinear eigenvalue problem:*

$$\left[ I - \beta C_{\mathrm{xx'}}^{\mathrm{dualIB}}(\hat{\mathbf{x}}, \beta) \right] \frac{\partial \log p_\beta(\mathbf{x'} \mid \hat{\mathbf{x}})}{\partial \hat{\mathbf{x}}} = 0 \,, \quad \left[ I - \beta C_{\mathrm{yy'}}^{\mathrm{dualIB}}(\hat{\mathbf{x}}, \beta) \right] \frac{\partial \log p_\beta(\mathbf{y'} \mid \hat{\mathbf{x}})}{\partial \hat{\mathbf{x}}} = 0, \quad (11)$$

*with the matrices $C^{\mathrm{dualIB}}$ given by:*

$$C_{\mathrm{xx'}}^{\mathrm{dualIB}}(\hat{\mathbf{x}};\beta) = \sum_{\mathbf{y},\tilde{\mathbf{y}},\tilde{\mathbf{x}}} p_\beta(\mathbf{y} \mid \hat{\mathbf{x}}) p_\beta(\tilde{\mathbf{x}} \mid \hat{\mathbf{x}}) \log \frac{p(\mathbf{y} \mid \mathbf{x})}{p(\mathbf{y} \mid \tilde{\mathbf{x}})} \cdot p_\beta(\mathbf{x'} \mid \hat{\mathbf{x}}) p_\beta(\tilde{\mathbf{y}} \mid \hat{\mathbf{x}}) \log \frac{p(\mathbf{y} \mid \mathbf{x'})}{p(\tilde{\mathbf{y}} \mid \mathbf{x'})}$$

$$C_{\mathrm{yy'}}^{\mathrm{dualIB}}(\hat{\mathbf{x}};\beta) = \sum_{\mathbf{x},\tilde{\mathbf{x}},\tilde{\mathbf{y}}} p_\beta(\mathbf{x} \mid \hat{\mathbf{x}}) p_\beta(\tilde{\mathbf{y}} \mid \hat{\mathbf{x}}) \log \frac{p(\mathbf{y} \mid \mathbf{x})}{p(\tilde{\mathbf{y}} \mid \mathbf{x})} \cdot p_\beta(\mathbf{y'} \mid \hat{\mathbf{x}}) p_\beta(\tilde{\mathbf{x}} \mid \hat{\mathbf{x}}) \log \frac{p(\mathbf{y'} \mid \mathbf{x})}{p(\mathbf{y'} \mid \tilde{\mathbf{x}})}. \quad (12)$$

The proof to *theorem* 2 is given in §A.3.

**Lemma 3.** *The matrices* $C_{xx'}^{\text{dualIB}}(\hat{x}; \beta), C_{yy'}^{\text{dualIB}}(\hat{x}; \beta)$ *have the same eigenvalues* $\{\lambda_i\}$*, with* $\lambda_1(\hat{x}) = 0$*. With binary* Y*, the critical points are obtained at* $\lambda_2(\hat{x}) = \beta^{-1}$*.*

The proof of *lemma* 3 given in section §A.3.1.

As in the IB, at the critical points, $\beta_c^{\text{dualIB}}$, the partial derivatives of the encoder and decoder with respect to $\beta$, $\partial \log p_\beta(x \mid \hat{x})/\partial\beta$, $\partial \log p_\beta(y \mid \hat{x})/\partial\beta$, have multiple (at least two) values. This results in discontinuities (cusps) in the encoder and decoder mutual information values as functions of $\beta$ along the optimal line, with an undefined second derivative.

## 2.2 THE INFORMATION PLANE OF THE dualIB

The *Information Plane*, $I_X = I(\hat{X}; X)$ vs. $I_Y = I(\hat{X}; Y)$, is the standard visualization of the compression-prediction trade off of the IB. It can be defined for *any* encoder once the decoder is the Bayes optimal (equation 3-$(iii)$), for which the $I_Y$ is the actual information of the representation on the desired label (Tishby et al., 1999).

Comparing the dualIB information curve to the IB curve shows the quality of this approximation. Figure 3(a) depicts this comparison. While we know that $I_Y^{\text{IB}}(\beta)$ is always higher, the two curves are almost indistinguishable. To better understand the relationship between these two curves, we look at the values of $I_X$ and $I_Y$ as functions of the corresponding $\beta$ (Figure 3(b),(c)). The important role of the critical points is revealed as the corresponding cusps along these curves. As we argue below, the IB information values are strictly below those of the dualIB, but the distance between them is minimized precisely at the dual critical points.

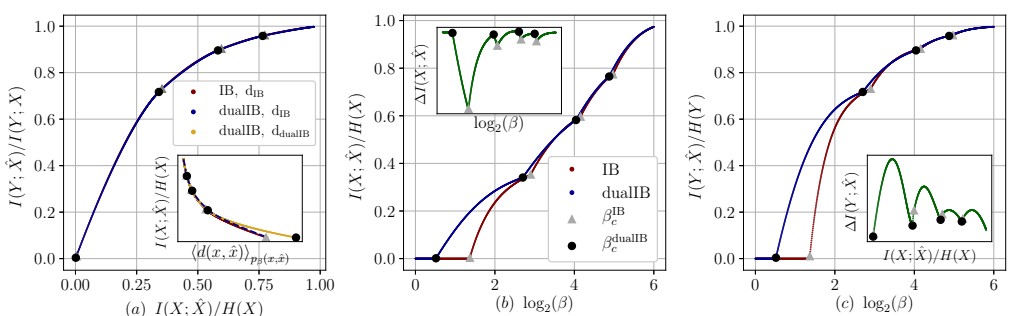

Figure 3: The IB's and dualIB's *Information Plane*. (a) $I_Y$ vs. $I_X$ for the two algorithms. The black dots are the dualIB critical points, $\beta_c^{\text{dualIB}}$, and the grey triangles are the IB critical points, $\beta_c^{\text{IB}}$. The corresponding distortion functions are shown in the inset. (b) The functions $I_X^{\text{IB}}(\beta)$ and $I_X^{\text{dualIB}}(\beta)$. Both curves are monotonic and concave between the critical points. The inset indicates the relative difference between the curves, where the alternating order of the critical points is clearly observed. (c) Similarly, $I_Y^{\text{IB}}(\beta)$ and $I_Y^{\text{dualIB}}(\beta)$ are monotonic and piece-wise concave. The relative discrepancy between the information curves is clearly minimized at the dualIB critical points (inset). The functions approach each other for large $\beta$.

**Lemma 4.** $I_X(\beta)$ *and* $I_Y(\beta)$*, along the optimal lines, are non-decreasing piece-wise concave, functions of* $\beta$*. When their second derivative (with respect to* $\beta$*) is defined, it is strictly negative.*

**Lemma 5.** *For any sub-optimal information curve* $(I_X, I_Y)$*,* $I_X^{\text{IB}}(\beta) \le I_X(\beta)$ *and* $I_Y^{\text{IB}}(\beta) \le I_Y(\beta)$*, for all values of* $\beta$*.*

Proofs of the above are given in §A.4.

The information plane properties are summarized by the following theorem.

**Theorem 6.** *(i) The critical points of the two algorithms alternate: for each critical point,* $\beta_c^{\text{dualIB}} \le \beta_c^{\text{IB}}$*. (ii) The distance between the two information curves is minimized precisely at the* dualIB *critical points* $\beta_c^{\text{dualIB}}$*. (iii) The two curves approach each other as* $\beta \to \infty$*.*

*Proof.* The proof follows from *lemmas* 4 and 5, together with the critical points analysis above, and is only sketched here. As the encoder and decoder at the critical points, $\beta_c^{\text{IB}}$ and $\beta_c^{\text{dualIB}}$, have

different left and right derivatives, they form cusps in the curves of the mutual information ($I_X$ and $I_Y$) as functions of $\beta$. These cusps can only be consistent with the optimality of the IB curves if $\beta_c^{\text{dualIB}} < \beta_c^{\text{IB}}$ (this is true for any sub-optimal distortion), otherwise the curves intersect.

Moreover, at the $\text{dualIB}$ critical points, the distance between the curves is minimized due to the strict concavity of the functions segments between the critical points. As the critical points imply discontinuity in the derivative, this results in a "jump" in the information values. Therefore, at any $\beta_c^{\text{dualIB}}$ the distance between the curves has a (local) minimum. This is depicted in Figure 3, comparing $I_X(\beta)$ and $I_Y(\beta)$ and their differences for the two algorithms.

The two curves approach each other for large $\beta$ since the two distortion functions become close in the low distortion limit (as long as $p(y \mid x)$ is bounded away from 0). □

## 3 THE EXPONENTIAL FAMILY dualIB

As discussed in §1, one of the major drawbacks of the IB is the extension of the decoder inputs from an empirical sample to new unseen patterns. To address this issue we can assume the existence of features, or statistics of the pattern, $A_r(x)$, which are sufficient or approximately sufficient for the prediction of the label Y. That is, we consider the rule's distribution, $p(y \mid x)$, to be an exponential family with these features. Exponential families form the elegant theoretical core of parametric statistics and often emerge as maximum entropy (Jaynes, 1957) or stochastic equilibrium distributions, subject to observed expectation constraints. They also form the class of parametric distributions for which exact, finite dimensional and additive, Minimal Sufficient Statistics exist (Kullback, 1959). Such distributions should also best match the rational behind the IB as extracting approximate sufficient statistics.

A key property of exponential families is their invariance under geometric averages. For this technical reason the dualIB encoder and decoder preserve the exponential form of the original data distribution for all values of $\beta$. Furthermore, the new (assumed limited to a finite dimension $d$) set of features/statistics are precisely the empirical expectations of these features. These $d$ empirical averages are the only numbers needed for predicting the label at the required accuracy, hence their advantage in reducing the encoder's computational complexity.

Assuming that the rule distribution is of the form:

$$p(y \mid x) = e^{-\sum_{r=0}^{d} \lambda^r(y) A_r(x)} = e^{-\boldsymbol{\lambda}(y) \cdot \boldsymbol{A}(x) - \log Z_{\mathbf{y}|\mathbf{x}}(x)} = \prod_{r=0}^{d} e^{-\lambda^r(y) A_r(x)} , \qquad (13)$$

where $A_r(x)$ are $d$ linearly independent functions of the input x and $\lambda^r(y)$ are functions of the label y, or the parameters of this exponential family. The $\boldsymbol{\lambda}(y)$ can also be considered Lagrange multipliers associated with the constraints conditional expectations $\mathbb{E}_{p(x|y)}[A_r(x)]$ in entropy maximization. The normalization factors, $Z_{\mathbf{y}|\mathbf{x}}(x)$, are written, for brevity, as $\lambda_{\mathbf{x}}^0 \equiv \log(\sum_{\mathbf{y}} \prod_{r=1}^{d} e^{-\lambda^r(y) A_r(x)})$ with $A_0(x) \equiv 1$. We do not constrain the marginal $p(x)$.

The important fact about the exponential form is that all the mutual information, $I(X; Y)$, is fully captured by the $d$ conditional expectations, $\mathbb{E}_{p(x|y)}[A_r(x)]$, since these are the (minimal) sufficient statistics for the parameters. This means that all the relevant information (in the training sample) is captured solely by $d$-dimensional empirical expectations. This can lead to a huge reduction in computational complexity (from $\dim(X)$ to $d$).

We next show that for the dualIB, this dimension reduction is preserved or improved along the dual information curve, for all values of $\beta$.

**Theorem 7.** *(dualExpIB) For data from an exponential family, equation 13, the optimal decoder of the* $\text{dualIB}$*, at a given $\beta$, is given by:*

$$p_\beta(y \mid \hat{x}) = e^{-\sum_{r=1}^{d} \lambda^r(y) A_{r,\beta}(\hat{x}) - \lambda_\beta^0(\hat{x})} , \quad \lambda_\beta^0(\hat{x}) = \log(\sum_y e^{-\sum_{r=1}^{d} \lambda^r(y) A_{r,\beta}(\hat{x})}) , \qquad (14)$$

*and the respective encoder:*

$$p_\beta(\hat{\mathbf{x}} \mid \mathbf{x}) = \frac{p_\beta(\hat{\mathbf{x}})e^{-\beta\lambda_\beta^0(\hat{\mathbf{x}})}}{Z_{\hat{\mathbf{x}}|\mathbf{x}}(\mathbf{x};\beta)}e^{-\beta\sum_{r=1}^d \lambda_\beta^r(\hat{\mathbf{x}})[A_r(\mathbf{x})-A_{r,\beta}(\hat{\mathbf{x}})]} \ , \tag{15}$$

*with the constraints and multipliers expectations,*

$$A_{r,\beta}(\hat{\mathbf{x}}) \equiv \sum_{\mathbf{x}} p_\beta(\mathbf{x} \mid \hat{\mathbf{x}})A_r(\mathbf{x}) \ , \ \lambda_\beta^r(\hat{\mathbf{x}}) \equiv \sum_{\mathbf{y}} p_\beta(\mathbf{y} \mid \hat{\mathbf{x}})\lambda^r(\mathbf{y}) \ , \ 1 \le r \le d \ . \tag{16}$$

The complete derivations for this section are given in §A.5.

This defines a simplified iterative algorithm to solve the dualExpIB problem, since we can replace the decoders' update rule at each iteration in the dualIB algorithm (Algorithm 1, row 6) with the simplified expression given in equation 14.

As a concrete illustration to the performance of the dualExpIB we examine a problem in which the rule distribution is given by an exponential family, $p(\mathbf{y} \mid \mathbf{x}) \sim \mathcal{N}(0, a + b|\sin(c\pi\mathbf{x})|)$, as in Globerson & Tishby (2003). Setting the constants $a, b, c$ we produce a sample $S$ (of size $N = 1000$) on which we calculate the empirical distribution, $\hat{p}(\mathbf{y} \mid \mathbf{x})$ and the empirical sufficient statistics $\hat{A}_r(\mathbf{x})$. Given the assumption on the rule we assess the exponential form empirical distribution, $\hat{p}_{\exp}(\mathbf{y} \mid \mathbf{x}; \hat{A}_r(\mathbf{x}))$. We then evaluate the IB on both empirical distributions, $\hat{p}$ and $\hat{p}_{\exp}$, and compare it to the dualExpIB given $\hat{A}_r(\mathbf{x}), \lambda^r(\mathbf{y})$[6].

The comparison between the IB and the dualExpIB, is given in Figure 4(a) as function of the continuous $\hat{y}$ for a fixed $\hat{x}$. It can be seen that the dualExpIB better captures the structure of the original data and is much less affected by the sampling noise. Next, we consider the *Information Plane* with respect to $\hat{Y}$, where the expected maximization by the dualExpIB is observed, Figure 4(b). To evaluate the prediction ability of the dualExpIB we consider the $\mathcal{L}_1$-*expected loss* with respect to the original distribution. Looking at Figure 4(c) it is apparent that for all values of the trade off parameter, $\beta$, the dualExpIB minimizes the loss providing the superiority in prediction.

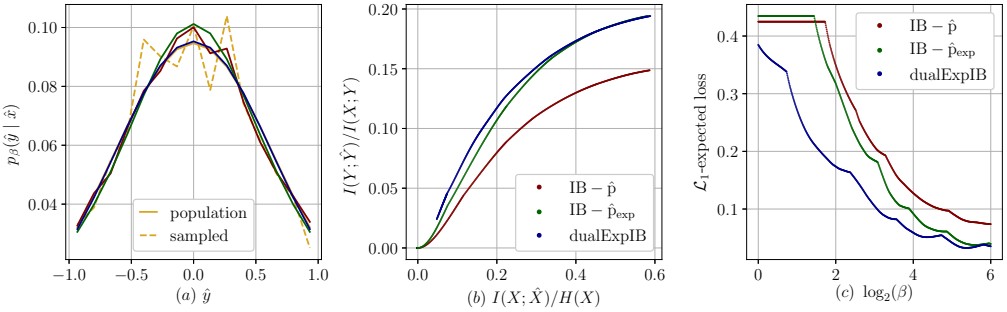

Figure 4: The IB's and dualExpIB's behavior over data sampled from $p(\mathbf{y} \mid \mathbf{x}) \sim \mathcal{N}(0, a + b|\sin(c\pi\mathbf{x})|)$ and $p(\mathbf{x}) \sim \mathcal{U}(-1, 1)$. (a) The decoder $p_\beta(\hat{y} \mid \hat{x})$ as a function of $\hat{y}$ for some $\hat{x} \in (-1, 1)$ at $\log_2(\beta) = 4$. (b) The *Information Plane* with respect to $\hat{Y}$, $I(Y; \hat{Y})$ vs. $I(X; \hat{X})$. (c) The $\mathcal{L}_1$-*expected loss*, $\mathbb{E}_{p(\mathbf{x})}[\|p(\hat{y} \mid \mathbf{x}) - p_\beta(\hat{y} \mid \mathbf{x})\|]$, as a function of the trade off parameter $\beta$.

## 4 CONCLUSION

We presented here the Dual Information Bottleneck (dualIB), a framework to preform the Information Bottleneck (IB) compression in a parametric setting. We have shown that the dualIB has several interesting properties: (i) it provides a good approximation to the original IB while using the low relevant dimensions of the original data. This can significantly reduce the complexity of finding good IB representations; (ii) it optimizes the information between the representation and the *predicted label* rather than the desired label as in the original IB. As we have shown, this can improve

---

[6]$\lambda^r(\mathbf{y})$ are defined by the specific exponential family.

the generalization error when trained on small samples since the predicted label is the one used in practice; (iii) It preserves the exponential form of data from exponential families, while reducing the dimensionality of the compressed representations. This important property was known to be satisfied by the Gaussian IB (Chechik et al., 2005) but not known for other distributions. The Gaussian case is self-dual in that sense. Generalizing this property to other exponential families was an open problem for many years; (iv) The exponential form of the optimal encoder-decoder pairs allows for the application to distributions with special symmetries, which can be naturally expressed in this form.

We conclude that the exponential Dual Information Bottleneck (dualExpIB) framework allows us now to approach problems previously considered intractable by the IB. Furthermore, it may allow better predictions to IB applications using finite samples or data with low internal dimensionality or special symmetries.

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

## A    APPENDIX

### A.1    DUALIB MATHEMATICAL FORMULATION

As presented in §1.6 the dualIB is solved with respect to the full Markov chain (equation 6) in which we introduce the new variable, $\hat{Y}$, the *predicted label*. Thus, in analogy to the IB we want to write the optimization problem in term of $\hat{Y}$.

Developing the expected distortion we find:

$$
\begin{aligned}
\mathbb{E}_{p_\beta(\mathbf{x},\hat{\mathbf{x}})}[d_{\mathrm{dualIB}}(\mathbf{x},\hat{\mathbf{x}})] &= \sum_{\mathbf{x},\hat{\mathbf{x}}} p_\beta(\mathbf{x},\hat{\mathbf{x}}) \sum_{\hat{\mathbf{y}}} p_\beta(\hat{\mathbf{y}} \mid \hat{\mathbf{x}}) \log \frac{p_\beta(\hat{\mathbf{y}} \mid \hat{\mathbf{x}})}{p(\hat{\mathbf{y}} \mid \mathbf{x})} \\
&= \sum_{\hat{\mathbf{x}},\hat{\mathbf{y}}} p_\beta(\hat{\mathbf{x}}) p_\beta(\hat{\mathbf{y}} \mid \hat{\mathbf{x}}) \log \frac{p_\beta(\hat{\mathbf{y}} \mid \hat{\mathbf{x}})}{p_\beta(\hat{\mathbf{y}})} - \sum_{\mathbf{x},\hat{\mathbf{y}}} p(\mathbf{x}) p_\beta(\hat{\mathbf{y}} \mid \mathbf{x}) \log \frac{p_\beta(\hat{\mathbf{y}} \mid \mathbf{x})}{p_\beta(\hat{\mathbf{y}})} \\
&\quad + \sum_{\mathbf{x},\hat{\mathbf{y}}} p(\mathbf{x}) p_\beta(\hat{\mathbf{y}} \mid \mathbf{x}) \log \frac{p_\beta(\hat{\mathbf{y}} \mid \mathbf{x})}{p(\hat{\mathbf{y}} \mid \mathbf{x})} \\
&= I(\hat{X};\hat{Y}) - I(X;\hat{Y}) + \mathbb{E}_{p(\mathbf{x})}[D[p_\beta(\hat{\mathbf{y}} \mid \mathbf{x}) \| p(\hat{\mathbf{y}} \mid \mathbf{x})]].
\end{aligned}
$$

Allowing the dual optimization problem to be written as:

$$
\mathcal{F}^*[p(\hat{\mathbf{x}} \mid \mathbf{x}); p(\hat{\mathbf{x}}); p(\mathbf{y} \mid \hat{\mathbf{x}})] = I(X;\hat{X}) - \beta \Big\{ I(X;\hat{Y}) - I(\hat{X};\hat{Y}) - \mathbb{E}_{p(\mathbf{x})}[D[p_\beta(\hat{\mathbf{y}} \mid \mathbf{x}) \| p(\hat{\mathbf{y}} \mid \mathbf{x})]] \Big\}.
$$

### A.2    THE DUALIB SOLUTIONS

To prove *theorem* 1 we want to obtain the normalized distributions minimizing the dualIB rate-distortion problem.

*Proof.* $(i)$ Given that the problem is formulated as a rate-distortion problem the encoder's update rule must be the known minimizer of the distortion function (Cover & Thomas, 2006). Thus the IB encoder with the dual distortion is plugged in. $(ii)$ For the decoder, by considering a small perturbation in the distortion $d_{\mathrm{dualIB}}(\mathbf{x},\hat{\mathbf{x}})$, with $\alpha(\hat{\mathbf{x}})$ the normalization Lagrange multiplier, we obtain:

$$
\delta d_{\mathrm{dualIB}}(\mathbf{x},\hat{\mathbf{x}}) = \delta \left( \sum_{\mathbf{y}} p_\beta(\mathbf{y} \mid \hat{\mathbf{x}}) \log \frac{p_\beta(\mathbf{y} \mid \hat{\mathbf{x}})}{p(\mathbf{y} \mid \mathbf{x})} + \alpha(\hat{\mathbf{x}}) \left( \sum_{\mathbf{y}} p_\beta(\mathbf{y} \mid \hat{\mathbf{x}}) - 1 \right) \right)
$$

$$
\frac{\delta d_{\mathrm{dualIB}}(\mathbf{x},\hat{\mathbf{x}})}{\delta p_\beta(\mathbf{y} \mid \hat{\mathbf{x}})} = \log \frac{p_\beta(\mathbf{y} \mid \hat{\mathbf{x}})}{p(\mathbf{y} \mid \mathbf{x})} + 1 + \alpha(\hat{\mathbf{x}}).
$$

Hence, minimizing the expected distortion becomes:

$$
\begin{aligned}
0 &= \sum_{\mathbf{x}} p_\beta(\mathbf{x} \mid \hat{\mathbf{x}}) \left[ \log \frac{p_\beta(\mathbf{y} \mid \hat{\mathbf{x}})}{p(\mathbf{y} \mid \mathbf{x})} + 1 \right] + \alpha(\hat{\mathbf{x}}) \\
&= \log p_\beta(\mathbf{y} \mid \hat{\mathbf{x}}) - \sum_{\mathbf{x}} p_\beta(\mathbf{x} \mid \hat{\mathbf{x}}) \log p(\mathbf{y} \mid \mathbf{x}) + 1 + \alpha(\hat{\mathbf{x}}),
\end{aligned}
$$

which yields Algorithm 1, row 6.    $\square$

Considering the dualIB encoder-decoder, Algorithm 1, we find that $\mathbb{E}_{p_\beta(\mathbf{x},\hat{\mathbf{x}})}[d_{\mathrm{dualIB}}(\mathbf{x},\hat{\mathbf{x}})]$ reduces to the expectation of the decoder's log partition function:

$$
\begin{aligned}
\mathbb{E}_{p_\beta(\mathbf{x},\hat{\mathbf{x}})}[d_{\mathrm{dualIB}}(\mathbf{x},\hat{\mathbf{x}})] &= \sum_{\mathbf{x},\hat{\mathbf{x}}} p_\beta(\mathbf{x},\hat{\mathbf{x}}) \sum_{\mathbf{y}} p_\beta(\mathbf{y} \mid \hat{\mathbf{x}}) \log \frac{p_\beta(\mathbf{y} \mid \hat{\mathbf{x}})}{p(\mathbf{y} \mid \mathbf{x})} \\
&= -\mathbb{E}_{p_\beta(\hat{\mathbf{x}})}\big[\log Z_{\mathbf{y}|\hat{\mathbf{x}}}(\hat{\mathbf{x}};\beta)\big] + \sum_{\hat{\mathbf{x}},\mathbf{y}} p_\beta(\hat{\mathbf{x}}) \left[ \sum_{\mathbf{x}'} p_\beta(\mathbf{x}' \mid \hat{\mathbf{x}}) \log p(\mathbf{y} \mid \mathbf{x}') - \sum_{\mathbf{x}} p_\beta(\mathbf{x} \mid \hat{\mathbf{x}}) \log p(\mathbf{y} \mid \mathbf{x}) \right] \\
&= -\mathbb{E}_{p_\beta(\hat{\mathbf{x}})}\big[\log Z_{\mathbf{y}|\hat{\mathbf{x}}}(\hat{\mathbf{x}};\beta)\big].
\end{aligned}
$$

### A.3 STABILITY ANALYSIS

Here we provide the detailed stability analysis allowing the definition of the matrices $C_{xx'}^{\text{dualIB}}, C_{yy'}^{\text{dualIB}}$ (equation 12) and which allows us to claim that they obey the same rules as the $C$ matrices (equation 5) in equation 4. Considering a variation in $\hat{x}$ we get:

$$
\frac{\partial \log p_\beta(x \mid \hat{x})}{\partial \hat{x}} = \beta \sum_y p_\beta(y \mid \hat{x}) \left( \log \frac{p(y \mid x)}{p_\beta(y \mid \hat{x})} - 1 \right) \frac{\partial \log p_\beta(y \mid \hat{x})}{\partial \hat{x}}
$$

$$
= \beta \sum_y p_\beta(y \mid \hat{x}) \left[ \log p(y \mid x) - \sum_{\tilde{x}} p_\beta(\tilde{x} \mid \hat{x}) \log p(y \mid \tilde{x}) \right] \frac{\partial \log p_\beta(y \mid \hat{x})}{\partial \hat{x}}
$$

$$
+ \beta \sum_y \log Z_{y \mid \hat{x}}(\hat{x}; \beta) \frac{\partial p_\beta(y \mid \hat{x})}{\partial \hat{x}}
$$

$$
= \beta \sum_{y, \tilde{x}} p_\beta(y \mid \hat{x}) p_\beta(\tilde{x} \mid \hat{x}) \log \frac{p(y \mid x)}{p(y \mid \tilde{x})} \frac{\partial \log p_\beta(y \mid \hat{x})}{\partial \hat{x}}, \tag{17}
$$

$$
\frac{\partial \log p_\beta(y \mid \hat{x})}{\partial \hat{x}} = - \frac{1}{Z_{y \mid \hat{x}}(\hat{x}; \beta)} \frac{\partial Z_{y \mid \hat{x}}(\hat{x}; \beta)}{\partial \hat{x}} + \sum_x p_\beta(x \mid \hat{x}) \log p(y \mid x) \frac{\partial \log p_\beta(x \mid \hat{x})}{\partial \hat{x}}
$$

$$
= - \sum_{\tilde{y}} p_\beta(\tilde{y} \mid \hat{x}) \sum_x p_\beta(x \mid \hat{x}) \log p(\tilde{y} \mid x) \frac{\partial \log p_\beta(x \mid \hat{x})}{\partial \hat{x}}
$$

$$
+ \sum_x p_\beta(x \mid \hat{x}) \log p(y \mid x) \frac{\partial \log p_\beta(x \mid \hat{x})}{\partial \hat{x}}
$$

$$
= \sum_{x, \tilde{y}} p_\beta(x \mid \hat{x}) p_\beta(\tilde{y} \mid \hat{x}) \log \frac{p(y \mid x)}{p(\tilde{y} \mid x)} \frac{\partial \log p_\beta(x \mid \hat{x})}{\partial \hat{x}}. \tag{18}
$$

Substituting equation 18 into equation 17 and vice versa one obtains:

$$
\frac{\partial \log p_\beta(x \mid \hat{x})}{\partial \hat{x}} = \beta \sum_{x', y, \tilde{y}, \tilde{x}} p_\beta(y \mid \hat{x}) p_\beta(\tilde{x} \mid \hat{x}) \log \frac{p(y \mid x)}{p(y \mid \tilde{x})}
$$

$$
\cdot p_\beta(x' \mid \hat{x}) p_\beta(\tilde{y} \mid \hat{x}) \log \frac{p(y \mid x')}{p(\tilde{y} \mid x')} \frac{\partial \log p_\beta(x' \mid \hat{x})}{\partial \hat{x}}
$$

$$
\frac{\partial \log p_\beta(y \mid \hat{x})}{\partial \hat{x}} = \beta \sum_{x, y', \tilde{x}, \tilde{y}} p_\beta(x \mid \hat{x}) p_\beta(\tilde{y} \mid \hat{x}) \log \frac{p(y \mid x)}{p(\tilde{y} \mid x)}
$$

$$
\cdot p_\beta(y' \mid \hat{x}) p_\beta(\tilde{x} \mid \hat{x}) \log \frac{p(y' \mid x)}{p(y' \mid \tilde{x})} \frac{\partial \log p_\beta(y' \mid \hat{x})}{\partial \hat{x}}.
$$

We now define the $C^{\text{dualIB}}$ matrices as follows:

$$
C_{xx'}^{\text{dualIB}}(\hat{x}; \beta) = \sum_{y, \tilde{y}, \tilde{x}} p_\beta(y \mid \hat{x}) p_\beta(\tilde{x} \mid \hat{x}) \log \frac{p(y \mid x)}{p(y \mid \tilde{x})} \cdot p_\beta(x' \mid \hat{x}) p_\beta(\tilde{y} \mid \hat{x}) \log \frac{p(y \mid x')}{p(\tilde{y} \mid x')}
$$

$$
C_{yy'}^{\text{dualIB}}(\hat{x}; \beta) = \sum_{x, \tilde{x}, \tilde{y}} p_\beta(x \mid \hat{x}) p_\beta(\tilde{y} \mid \hat{x}) \log \frac{p(y \mid x)}{p(\tilde{y} \mid x)} \cdot p_\beta(y' \mid \hat{x}) p_\beta(\tilde{x} \mid \hat{x}) \log \frac{p(y' \mid x)}{p(y' \mid \tilde{x})}.
$$

Using the above definition we have an equivalence to equation 4 in the form of:

$$
\left[ I - \beta C_{xx'}^{\text{dualIB}}(\hat{x}, \beta) \right] \frac{\partial \log p_\beta(x' \mid \hat{x})}{\partial \hat{x}} = 0, \quad \left[ I - \beta C_{yy'}^{\text{dualIB}}(\hat{x}, \beta) \right] \frac{\partial \log p_\beta(y' \mid \hat{x})}{\partial \hat{x}} = 0.
$$

Note that for the binary case, the matrices may be simplified to:

$$
C_{xx'}^{\text{dualIB}}(\hat{x}; \beta) = \sum_{y, \tilde{x}} p_\beta(y \mid \hat{x}) p_\beta(\tilde{x} \mid \hat{x}) \log \frac{p(y \mid x)}{p(y \mid \tilde{x})} \cdot p_\beta(x' \mid \hat{x})(1 - p_\beta(y \mid \hat{x})) \log \frac{p(y \mid x')}{1 - p(y \mid x')}
$$

$$
C_{yy'}^{\text{dualIB}}(\hat{x}; \beta) = \sum_{x, \tilde{x}} p_\beta(x \mid \hat{x})(1 - p_\beta(y \mid \hat{x})) \log \frac{p(y \mid x)}{1 - p(y \mid x)} \cdot p_\beta(y' \mid \hat{x}) p_\beta(\tilde{x} \mid \hat{x}) \log \frac{p(y' \mid x)}{p(y' \mid \tilde{x})}.
$$

### A.3.1 PROOF OF LEMMA 3

We show that the $C^{\text{dualIB}}$ matrices share the same eigenvalues with $\lambda_1(\hat{x}) = 0$.

*Proof.* The matrices, $C_{xx'}^{\text{dualIB}}(\hat{x};\beta)$, $C_{yy'}^{\text{dualIB}}(\hat{x};\beta)$, are given by:

$$C_{xx'}^{\text{dualIB}}(\hat{x};\beta) = A_{xy}(\hat{x};\beta)B_{yx'}(\hat{x};\beta) \ , \ C_{yy'}^{\text{dualIB}}(\hat{x};\beta) = B_{yx}(\hat{x};\beta)A_{xy'}(\hat{x};\beta),$$

with:

$$A_{xy}(\hat{x};\beta) = p_\beta(y\mid\hat{x})\sum_{\tilde{x}}p_\beta(\tilde{x}\mid\hat{x})\log\frac{p(y\mid x)}{p(y\mid\tilde{x})} \ , \ B_{yx}(\hat{x};\beta) = p_\beta(x\mid\hat{x})\sum_{\tilde{y}}p_\beta(\tilde{y}\mid\hat{x})\log\frac{p(y\mid x)}{p(\tilde{y}\mid x)}.$$

Given that the matrices are obtained by the multiplication of the same matrices, it follows that they have the same eigenvalues $\{\lambda_i(\hat{x};\beta)\}$.

To prove that $\lambda_1(\hat{x};\beta) = 0$ we show that $\det(C_{yy'}^{\text{dualIB}}) = 0$. We present the exact calculation for a binary label $Y \in \{y_0, y_1\}$ (the argument for general Y follows by encoding the label as a sequence of bits and discussing the first bit only, as a binary case):

$$\det(C_{yy'}^{\text{dualIB}}(\hat{x};\beta)) = \sum_{x,\tilde{x}}p_\beta(x\mid\hat{x})p_\beta(y_1\mid\hat{x})\log\frac{p(y_0\mid x)}{p(y_1\mid x)}\cdot p_\beta(y_0\mid\hat{x})p_\beta(\tilde{x}\mid\hat{x})\log\frac{p(y_0\mid x)}{p(y_0\mid\tilde{x})}$$

$$\cdot\sum_{x',\tilde{x}',}p_\beta(x'\mid\hat{x})p_\beta(y_0\mid\hat{x})\log\frac{p(y_1\mid x')}{p(y_0\mid x')}\cdot p_\beta(y_1\mid\hat{x})p_\beta(\tilde{x}'\mid\hat{x})\log\frac{p(y_1\mid x')}{p(y_1\mid\tilde{x}')}$$

$$-\sum_{x,\tilde{x}}p_\beta(x\mid\hat{x})p_\beta(y_0\mid\hat{x})\log\frac{p(y_1\mid x)}{p(y_0\mid x)}\cdot p_\beta(y_0\mid\hat{x})p_\beta(\tilde{x}\mid\hat{x})\log\frac{p(y_0\mid x)}{p(y_0\mid\tilde{x})}$$

$$\cdot\sum_{x',\tilde{x}'}p_\beta(x'\mid\hat{x})p_\beta(y_1\mid\hat{x})\log\frac{p(y_0\mid x')}{p(y_1\mid x')}\cdot p_\beta(y_1\mid\hat{x})p_\beta(\tilde{x}'\mid\hat{x})\log\frac{p(y_1\mid x')}{p(y_1\mid\tilde{x}')}$$

$$= \sum_{x,x',\tilde{x},\tilde{x}'}p_\beta(x\mid\hat{x})p_\beta(x'\mid\hat{x})p_\beta^2(y_0\mid\hat{x})p_\beta^2(y_1\mid\hat{x})p_\beta(\tilde{x}\mid\hat{x})\log\frac{p(y_0\mid x)}{p(y_0\mid\tilde{x})}p_\beta(\tilde{x}'\mid\hat{x})\log\frac{p(y_1\mid x')}{p(y_1\mid\tilde{x}')}$$

$$\cdot\left[\log\frac{p(y_0\mid x)}{p(y_1\mid x)}\log\frac{p(y_1\mid x')}{p(y_0\mid x')}-\log\frac{p(y_0\mid x)}{p(y_1\mid x)}\log\frac{p(y_1\mid x')}{p(y_0\mid x')}\right] = 0.$$

Given that the determinant is 0 implies that $\lambda_1(\hat{x}) = 0$. $\square$

For a binary problem we can describe the non-zero eigenvalue using $\lambda_2(\hat{x}) = \text{Tr}(C_{yy'}^{\text{dualIB}}(\hat{x};\beta))$. That is:

$$\lambda_2(\hat{x}) = \sum_{x,\tilde{x}}p_\beta(x\mid\hat{x})p_\beta(y_1\mid\hat{x})\log\frac{p(y_0\mid x)}{p(y_1\mid x)}\cdot p_\beta(y_0\mid\hat{x})p_\beta(\tilde{x}\mid\hat{x})\log\frac{p(y_0\mid x)}{p(y_0\mid\tilde{x})}$$

$$+\sum_{x,\tilde{x}}p_\beta(x\mid\hat{x})p_\beta(y_0\mid\hat{x})\log\frac{p(y_1\mid x)}{p(y_0\mid x)}\cdot p_\beta(y_1\mid\hat{x})p_\beta(\tilde{x}\mid\hat{x})\log\frac{p(y_1\mid x)}{p(y_1\mid\tilde{x})}$$

$$= p_\beta(y_1\mid\hat{x})p_\beta(y_0\mid\hat{x})\sum_{x,\tilde{x}}p_\beta(x\mid\hat{x})p_\beta(\tilde{x}\mid\hat{x})\log\frac{p(y_0\mid x)}{p(y_1\mid x)}\left[\log\frac{p(y_0\mid x)}{p(y_0\mid\tilde{x})}-\log\frac{p(y_1\mid x)}{p(y_1\mid\tilde{x})}\right].$$

### A.4 INFORMATION PLANE ANALYSIS

We rely on known results for the rate-distortion problem and the information plane:

**Lemma 8.** $I(X;\hat{X})$ *is a non-increasing convex function of the distortion* $\mathbb{E}_{p_\beta(x,\hat{x})}[d(x,\hat{x})]$ *with a slope of* $-\beta$.

We emphasis that this is a general result of rate-distortion thus holds for the dualIB as well.

**Lemma 9.** *For a fixed encoder $p_\beta(\hat{x} \mid x)$ and the Bayes optimal decoder $p_\beta(y \mid \hat{x})$:*

$$\mathbb{E}[d_{\text{IB}}(x, \hat{x})] = I(X; Y) - I(\hat{X}; Y).$$

*Thus, the information curve, $I_Y$ vs. $I_X$, is a non-decreasing concave function with a positive slope, $\beta^{-1}$. The concavity implies that $\beta$ increases along the curve.*

(Cover & Thomas, 2006; Gilad-bachrach et al., 2003).

### A.4.1 PROOF OF LEMMA 4

In the following section we provide a proof to *lemma* 4, for the IB and dualIB problems.

*Proof.* We want to analyze the behavior of $I_X(\beta), I_Y(\beta)$, that is the change in each term as a function of the corresponding $\beta$. From *lemma* 9, the concavity of the information curve, we can deduce that both are non-decreasing functions of $\beta$. As the two $\beta$ derivatives are proportional it's enough to discuss the first one.

Next, we focus on their behavior between two critical points. That is, where the cardinality of $\hat{X}$ is fixed (clusters are "static"). For "static" clusters, the $\beta$ derivative of $I_X$, along the optimal line is given by:

$$\begin{aligned}
\frac{\partial I(X; \hat{X})}{\partial \beta} &= -\frac{\partial}{\partial \beta}\left[\sum_{x,\hat{x}} p_\beta(x, \hat{x})\big(\log Z_{\hat{x}|x}(x; \beta) + \beta d(x, \hat{x})\big)\right] \\
&= -\beta\left\langle d(x, \hat{x})\frac{\partial \log p_\beta(\hat{x} \mid x)}{\partial \beta}\right\rangle_{p_\beta(x, \hat{x})} \\
&\approx \beta\left\langle d(x, \hat{x})\left[\frac{\partial \log Z_{\hat{x}|x}(x; \beta)}{\partial \beta} + d(x, \hat{x})\right]\right\rangle_{p_\beta(x, \hat{x})} \\
&\approx \beta\left\langle \underbrace{\left\langle d^2(x, \hat{x})\right\rangle_{p_\beta(\hat{x}|x)} - \left\langle d(x, \hat{x})\right\rangle^2_{p_\beta(\hat{x}|x)}}_{\text{Var}(d(x))}\right\rangle_{p(x)}.
\end{aligned}$$

This first of all reassures that the function is non-decreasing as $\text{Var}(d(x)) \geq 0$.

The piece-wise concavity follows from the fact that when the number of clusters is fixed (between the critical points) - increasing $\beta$ decreases the clusters conditional entropy $H(\hat{X} \mid X)$, as the encoder becomes more deterministic. The mutual information is bounded by $H(\hat{X})$ and it's $\beta$ derivative decreases. Further, between the critical points there are no sign changes in the second $\beta$ derivative. $\square$

### A.4.2 PROOF OF LEMMA 5

*Proof.* The information curve has a positive slope, $\beta^{-1}$, with $\beta$ increasing along it, *lemma* 9. That is, given a value of $\beta$, there exists a pair $I_Y^{\text{IB}}(\beta), I_X^{\text{IB}}(\beta)$ such that $\partial I_Y^{\text{IB}}(\beta)/\partial I_X^{\text{IB}}(\beta) = \beta^{-1}$. Now, consider a sub-optimal information curve, $I_Y^*, I_X^*$. There exist values $\beta', \beta''$ such that:

$$I_X^*(\beta') = I_X^{\text{IB}}(\beta), \quad I_Y^*(\beta'') = I_Y^{\text{IB}}(\beta).$$

The optimality of the IB implies that sub-optimal curves lie below it; i.e, the IB slope is steeper:

$$\beta^{-1} > \beta'^{-1}, \quad \beta^{-1} > \beta''^{-1}.$$

Thus, given that $\beta$ increases along the information curve, it holds that:

$$I_X^*(\beta) > I_X^*(\beta') = I_X^{\text{IB}}(\beta), \quad I_Y^*(\beta) > I_Y^*(\beta'') = I_Y^{\text{IB}}(\beta).$$

$\square$

## A.5 DERIVATION OF THE DUALEXPIB

We provide elaborate derivations to *theorem* 7; that is, we obtain the dualIB optimal encoder-decoder under the exponential assumption over the data. We use the notations defined in §3.

- The *decoder*, equation 14.
  Substituting the exponential assumption into the dualIB log-decoder yields:

$$
\log p_\beta(\mathbf{y} \mid \hat{\mathbf{x}}) = \sum_{\mathbf{x}} p_\beta(\mathbf{x} \mid \hat{\mathbf{x}}) \log p(\mathbf{y} \mid \mathbf{x}) - \log Z_{\mathbf{y}|\hat{\mathbf{x}}}(\hat{\mathbf{x}}; \beta)
$$

$$
= - \sum_{\mathbf{x}} \sum_{r=0}^{d} p_\beta(\mathbf{x} \mid \hat{\mathbf{x}}) \lambda^r(\mathbf{y}) A_r(\mathbf{x}) - \log Z_{\mathbf{y}|\hat{\mathbf{x}}}(\hat{\mathbf{x}}; \beta)
$$

$$
= - \sum_{r=1}^{d} \lambda^r(\mathbf{y}) A_{r,\beta}(\hat{\mathbf{x}}) - \mathbb{E}_{p_\beta(\mathbf{x}|\hat{\mathbf{x}})}\left[\lambda_{\mathbf{x}}^0\right] - \log Z_{\mathbf{y}|\hat{\mathbf{x}}}(\hat{\mathbf{x}}; \beta).
$$

  Taking a closer look at the normalization term:

$$
Z_{\mathbf{y}|\hat{\mathbf{x}}}(\hat{\mathbf{x}}; \beta) = \sum_{\mathbf{y}} e^{\sum_{\mathbf{x}} p_\beta(\mathbf{x}|\hat{\mathbf{x}}) \log p(\mathbf{y}|\mathbf{x})} = e^{-\mathbb{E}_{p_\beta(\mathbf{x}|\hat{\mathbf{x}})}\left[\lambda_{\mathbf{x}}^0\right]} \sum_{\mathbf{y}} e^{-\sum_{r=1}^{d} \lambda^r(\mathbf{y}) A_{r,\beta}(\hat{\mathbf{x}})}
$$

$$
\log Z_{\mathbf{y}|\hat{\mathbf{x}}}(\hat{\mathbf{x}}; \beta) = -\mathbb{E}_{p_\beta(\mathbf{x}|\hat{\mathbf{x}})}\left[\lambda_{\mathbf{x}}^0\right] + \log\left(\sum_{\mathbf{y}} e^{-\sum_{r=1}^{d} \lambda^r(\mathbf{y}) A_{r,\beta}(\hat{\mathbf{x}})}\right).
$$

  From which it follows that $\lambda_\beta^0(\hat{\mathbf{x}})$ is given by:

$$
\lambda_\beta^0(\hat{\mathbf{x}}) = \log\left(\sum_{\mathbf{y}} e^{-\sum_{r=1}^{d} \lambda^r(\mathbf{y}) A_{r,\beta}(\hat{\mathbf{x}})}\right),
$$

  and we can conclude that the dualExpIB decoder takes the form:

$$
\log p_\beta(\mathbf{y} \mid \hat{\mathbf{x}}) = - \sum_{r=0}^{d} \lambda^r(\mathbf{y}) A_{r,\beta}(\hat{\mathbf{x}}).
$$

- The *encoder*, equation 15.
  The core of the encoder is the dual distortion function which may now be written as:

$$
d_{\text{dualIB}}(\mathbf{x}, \hat{\mathbf{x}}) = \sum_{\mathbf{y}} p_\beta(\mathbf{y} \mid \hat{\mathbf{x}}) \log \frac{p_\beta(\mathbf{y} \mid \hat{\mathbf{x}})}{p(\mathbf{y} \mid \mathbf{x})}
$$

$$
= \sum_{\mathbf{y}} p_\beta(\mathbf{y} \mid \hat{\mathbf{x}}) \left[ \left(\lambda_{\mathbf{x}}^0 - \lambda_\beta^0(\hat{\mathbf{x}})\right) + \sum_{r=1}^{d} \lambda^r(\mathbf{y})(A_r(\mathbf{x}) - A_{r,\beta}(\hat{\mathbf{x}})) \right]
$$

$$
= \lambda_{\mathbf{x}}^0 - \lambda_\beta^0(\hat{\mathbf{x}}) + \sum_{r=1}^{d} \lambda_\beta^r(\hat{\mathbf{x}})(A_r(\mathbf{x}) - A_{r,\beta}(\hat{\mathbf{x}})),
$$

  substituting this into the encoder's definition we obtain:

$$
p_\beta(\hat{\mathbf{x}} \mid \mathbf{x}) = \frac{p_\beta(\hat{\mathbf{x}})}{Z_{\hat{\mathbf{x}}|\mathbf{x}}(\mathbf{x}; \beta)} e^{-\beta\left[\lambda_{\mathbf{x}}^0 - \lambda_\beta^0(\hat{\mathbf{x}}) + \sum_{r=1}^{d} \lambda_\beta^r(\hat{\mathbf{x}})[A_r(\mathbf{x}) - A_{r,\beta}(\hat{\mathbf{x}})]\right]}
$$

$$
= \frac{p_\beta(\hat{\mathbf{x}}) e^{\beta \lambda_\beta^0(\hat{\mathbf{x}})}}{Z_{\hat{\mathbf{x}}|\mathbf{x}}(\mathbf{x}; \beta)} e^{-\beta \sum_{r=1}^{d} \lambda_\beta^r(\hat{\mathbf{x}})[A_r(\mathbf{x}) - A_{r,\beta}(\hat{\mathbf{x}})]}.
$$

