# OpenReview forum: "The Dual Information Bottleneck"
_ICLR.cc/2020/Conference — Reject_

### Official Review · AnonReviewer3 · 2019-10-23
**Official Blind Review #3**

**Rating:** 6

**Review:**

This paper proposes a new "dual" variant of the Information Bottleneck framework. The IB framework has been the subject of many papers in past years, with a focus on understanding the inner workings of deep learning. The framework poses machine learning as optimizing an internal representation to tradeoff between retaining less information about the input features and retaining more information about the output label (prediction). The existing framework measures the retained information about the prediction via mutual information, which can be expressed as a KL divergence. The new dual framework reverses the arguments of this divergence.

The paper shows that this dual IB closely mirrors the original IB while having several additional nice mathematical properties. In particular,  it can be shown that for exponential families the dual IB representation retains the exponential form.

Overall, I think this paper adds a meaningful new perspective to the IB framework and the analysis appears to be thorough. As such, I support acceptance.

The paper could be improved by giving more interpretation of the formal results and experiments - i.e. tying this framework back to the higher-level questions and explaining what the quantities mean.

Figures 1, 2 part a: The meaning of both axes is unclear. The horizontal axis beta is a lagrangian parameter with no meaning outside the framework. The vertical axis Pr[y=0|\hat x] has no semantics since the problem has not been defined. What is the reader meant to take away from these figures?

Equation 3, part i: The numerator should have a beta subscript. The meaning of the denominator is not clear (it is a normalizing constant).



**Experience Assessment:**

I have read many papers in this area.

**Review Assessment: Checking Correctness Of Derivations And Theory:**

I assessed the sensibility of the derivations and theory.

**Review Assessment: Checking Correctness Of Experiments:**

I did not assess the experiments.

**Review Assessment: Thoroughness In Paper Reading:**

I read the paper thoroughly.

---

> ### Author Response · Authors · 2019-11-15
> **Response to Review #3**
>
> We thank the reviewer for the helpful comments. We were delighted to see that the reviewer agrees with our understanding of the contribution and innovations that the formalism suggest in comparison to the known $\rm{IB}$.
>
> We will now relate to specific points raised be the reviewer:
>
> - Motivation:  We refer the reviewer to the general official comment posted.
>
> - Interpretation of the results:  Following the reviewers comment we have added examples to previous application of the$\rm{IB}$  (sec. 1.1 “The Information Bottleneck method”) to tie the general framework into context and believe that now the dualIB framework can be tested and applied to on similar tasks.
>
> - Figures explanation:  In addition we understood that the text was lacking an explanation of the experiments and the interpretations of the results, Thus we added an explanation regarding the results and settings for the problem presented in figure 1(a),2(a).
>
> - We thank the reviewer for pointing out the typo at Equation 3 (fixed in the newly submitted version).

---

### Official Review · AnonReviewer1 · 2019-10-24
**Official Blind Review #1**

**Rating:** 3

**Review:**

This paper introduces a variant of the Information Bottleneck (IB) framework, which consists in permuting the conditional probabilities of y given x and y given \hat{x} in a Kullback-Liebler divergence involved in the IB optimization criterion.

Interestingly, this change only results in changing an arithmetic mean into a geometric mean in the algorithmic resolution.

Good properties of the exponential families (existence of non-trivial minimal sufficient statistics) are preserved, and an analysis of the new critical points/information plane induced is carried out.

The paper is globally well written and clear, and the maths are rigorously introduced and treated. Two minor comments would concern the definition of the Mutual Information (MI), which could be recalled to help the unfamiliar reader and improve self-containedness, and the notation of the expected value (\langle \rangle_p), unusual in Machine Learning where \mathbb{E} is often preferred.

Another point that could be enhanced is the intuition behind the IB and dualIB criteria: a small comment on their meaning/relevance as well as the rationale/implication of the probabilities permutation would be valuable addition.

Exhibiting a link with variational autoencoders (VAEs), and expliciting the differences in a VAE framework between the two criteria could also represent an interesting parallel, especially for machine learning oriented readers.

One of the main drawback of the present paper is however the lack of convincing experiments. Graphs showing the good behavior of the introduced framework are ok, but the clear interest of using dualIB rather than IB could be emphasized more. In particular, it does not seem that the issues about IB raised in the abstract (curse of dimensionality, no closed form solution) are solved with dualIB. Furthermore, dualIB optimizes the MI of the predicted labels, which is claimed to be beneficial contrary to the MI on the actual labels. However, no empirical demonstration of this superiority is produced, which is a bit disappointing.

**Experience Assessment:**

I have published one or two papers in this area.

**Review Assessment: Checking Correctness Of Derivations And Theory:**

I carefully checked the derivations and theory.

**Review Assessment: Checking Correctness Of Experiments:**

I carefully checked the experiments.

**Review Assessment: Thoroughness In Paper Reading:**

I read the paper thoroughly.

---

> ### Author Response · Authors · 2019-11-15
> **Response to Review #1**
>
> We were glad to see that the conceptual idea along with the innovation presented in the Dual Information Bottleneck ($\rm{dualIB}$) framework was well understood.
> We thank the reviewer for raising critical points we failed to address and/or properly explain, we have revised the paper accordingly as elaborated in the general official comment posted and will now relate to specific points raised by the reviewer:
>
> - Motivation: We refer the reviewer to the general official comment posted.
>
> - Definitions and notations: Following the reviewer's remark we have added more background defining the Mutual Information (sec. 1.1 “The Information Bottleneck method”) and adapted the notations of the expected value to the ML preferred formalism ($\mathbb{E}$).
>
> - The $\rm{dualIB}$form: The reviewer's comment has brought to our attention that the rationale behind the obtained dual form, that is the permutation of the probabilities distribution in the rate distortion function was not made clear in the text, we have added an explanation in the text (sec. 3 “The Exponential Family dualIB”) and would like to elaborate it here:
> Given the above motivation, we looked for the minimization problem preserving the structure on the decoder, $p(y | \hat{x})$, an exponential form (not the Bayes optimal we get by minimizing the IB Lagrangian).
> A natural alteration achieving this goal is the geometric mean (rather than an arithmetic one) from which it follows that the distortion function minimizing it is the geometric dual distortion; the KL divergence given by the permutation of the distributions.
>
> - VAEs: We agree that this could be interesting but unfortunately within the scope of the paper we could not include this analysis.
>
> - New empirical results: While this work is mainly theoretical and intends to lay down the formalism of the $\rm{dualIB}$ framework we agree that an empirical result could better convey it’s advantages and thus we added a new experiment in sec. 3 “The Exponential Family dualIB”:
>
> In the new experiment we apply the $\rm{IB}$ and $\rm{dualExpIB}$ to data sampled from a Normal distribution.
> The visual results, given in Figure 4(a) now showcase the benefits and improvements of the framework in this scenario.
> In Figure 4(a) we provide examples of the predicted distributions in which one can see the ability of the $\rm{dualExpIB}$ to better generalize and capture the original distribution even though it is given the empirical sufficient statistics of the sample.
> Next, we aim to visualize the superiority in prediction; we wish clarify that this in this we referred to the maximization of the information on the newly added prediction variable, $\hat{Y}$, as it rises naturally from the new Lagrangian.
> In Figure 4(b) we show the $\hat{Y}$’s information plane it for the new example. The superiority of $\rm{dualExpIB}$ over the $\rm{IB}$  is apparent.
> Furthermore, we evaluate the prediction ability over the examined cases using the $\mathcal{L}_1$-expected loss, testing the accuracy between the obtained predictions of each method to the original population distribution, see Figure 4(c).
> We see that for all values of the trade off parameter, $\beta$, the loss is minimized by the $\rm{dualExpIB}$.
>
> To conclude, We hope that we were able to portray the advantages and interest in the $\rm{dualIB}$ framework.
> The framework is given from a theoretical point of view but we hope that with a clearer view of the motivation along with visual examples and elaboration on possible applications, it’s importance and relevance to the ICLR conference is apparent.

---

### Official Review · AnonReviewer2 · 2019-11-10
**Official Blind Review #2**

**Rating:** 3

**Review:**

Description:

The information bottleneck (IB) is an information theoretic principle for optimizing a mapping (encoding, e.g. clustering) of an input, to trade off two kinds of mutual information: minimize mutual information between the original input and the mapped version (to compress the input), and maximize mutual information between the mapped input and an output variable. It is related to a minimization of an (expected) Kullback-Leibler divergence betwen conditional distributions of an output variable.

In this paper, instead of the original IB, authors consider a previously presented dual problem of Felice and Ay, where the Kullback-Leibler divergence is minimized in the reverse direction: from the conditional distribution of output given encoding, p(y|hat x), to the conditional distribution given the original input, p(y|x).
The "dual problem" itself has a more complicated form than the original IB, authors claim this is "a good approximation" of the original bottleneck formulation, and aim to prove various "interesting properties" of it.

- An iterative algorithm (Algorithm 1) similar to the original IB algorithm but with a few more steps is provided.

- A theorem about critical points where cardinality of the representation changes is given, similar to the IB critical points, and another theorem about difference of curves on an information plane between the IB and dual-IB solutions.

- Authors also show that if the true conditional distribution of outputs given inputs has an exponential-family form, the dual-IB decoder also has a form in the same family, which is said to reduce computational complexity of the algorithm.



Evaluation:

This is an entirely theory-based paper; although an algorithm is given, it is not instantiated for any concrete representation learning task, and no experiments at all are demonstrated.

Overall, I feel the motivation is not clear and strong enough. The abstract does not illustrate the importance of the mentioned "interesting properties" well enough for concrete tasks. Reading the paper, the clearest motivations seem to be improving computational complexity, and having a clearer connection to output prediction in cases where the predictor may be sub-optimal. However, authors do not quantify these well:

- The computational complexity improvement is not made clear (quantified) in a concrete IB optimization task: it seems it is only for exponential families, and even for them only affects one part of the algorithm, reducing its complexity from dim(X) to d; the impact of this is not tried out in any experiment.

- For output prediction, authors motivate that dual-IB could have a more direct connection e.g. "due to finite sample, in which it can be very different from the one obtained from the full distribution"). Authors further claim that the dual-IB formulation can "improve the generalization error when trained on small samples since the predicted label is the one used in practice". However, this is not tested at all: no prediction experiments, no quantification of generalization error, and no comparisons are done, thus the impact of the clearer connection to output prediction is not tested at all, and no clear theorems are given about it either.

The property that the algorithm "preserves exponential form of the original data distribution, if one exists" is interesting in principle, but it is unclear if any real data would anyway precisely have such a distribution; what happens if the data is not exactly in an exponential family?

In its current state the paper, although based on an interesting direction, in my opinion does not make a sufficient impact to be accepted to ICLR.

Other comments:

"Application to deep learning" mentioned in Section 1.5 is only a sincle sentence in the conclusions.

There have been some other suggested alternative IB formulations, for example the Deterministic IB of Strouse and Schwab (Neural Computation 2017) which also claim improved computational efficiency. How does the method compare to those?

Section 1.5 claims the algorithm "preserves the low dimensional sufficient statistics of the data": it is not clear what "preserves" means here, certainly it seems the decoder in Theorem 7 uses the same kinds of sufficient statistics as in the original data, but it is not clear that hat(x) would somehow preserve the same values of the sufficient statistics.


**Experience Assessment:**

I have published one or two papers in this area.

**Review Assessment: Checking Correctness Of Derivations And Theory:**

I did not assess the derivations or theory.

**Review Assessment: Checking Correctness Of Experiments:**

N/A

**Review Assessment: Thoroughness In Paper Reading:**

I made a quick assessment of this paper.

---

> ### Author Response · Authors · 2019-11-15
> **Response to Review #2**
>
> We wish to thank to reviewer for his response, while we stand behind it being a theoretical paper (a point we soon relate to in detail) we agree that the paper did not convey the motivation behind the new framework presented and with that the advantages it has. We have revised the paper accordingly as elaborated in the general official comment posted.
>
> We will now relate to specific points raised be the reviewer:
>
> - Theory-based paper:  We agree with the reviewer that this paper is theory-based as it aims to reveal a new framework, the Dual Information Bottleneck ($\rm{dualIB}$), and lay down the grounds for future applications of it. As such we find it necessary to analyse it’s analytical/theoretical properties and compare it to the original form, the Information Bottleneck ($\rm{lIB}$).
> Having said that, we understand the need of empirical experiments to demonstrate the claims and have thus added a new example that is analysed in the text, see sec. 3 “The Exponential Family dualIB”. This is still not an application of a concrete representation learning task yet it provides a visual evidence to the claimed behaviour.
>
> - Motivation:  We refer the reviewer to the general official comment posted.
>
> - Computational complexity:  The computation complexity reduction emerges from the analytical result (Theorem 7) in which we show that the encoder-decoder are given by the sufficient statistics of the input distribution. Thus, the task is now to track these rather than the full distribution. This result is appealing to many problems which have a natural parametrization (i.e  a physical problem defined by a given Hamiltonian thus a Gibbs distribution) but are intractable using the IB.
>
> - Output prediction: We clarify that in this we refer to the maximization of the information on the newly added prediction variable, $\hat{Y}$, as it rises naturally from the new Lagrangian. Thus the theory behind this lays in the formalism of the problem. in the revised version a visualization of the superiority is shown for the new setting and the phenomenon is clearly, see Figure 4(b).
> As suggested by the reviewer we further evaluate now the prediction ability over the examined cases by displaying the $\mathcal{L}_1$-expected loss which makes it evident that the $\rm{dualExpIB}$ outperforms the IB minimizing the loss for all values of the trade off parameter, see Figure 4(c).
>
> - Data from exponential families or “almost”:  We thank the reviewer for pointing out that the motivation for choosing the exponential families was unclear and made an attempt to clarify this in the text (see Sec. 3 “The Exponential Family dualIB” ). From a statistical point of view a natural way of parametrizing data is obtained by a distribution from the exponential family as most of the commonly used distributions form an exponential family or subset of an exponential family.
> Moreover the exponential family form emerges when asking what is the maximum-entropy distribution consistent with given constraints on expected values.
> Within the scope of the paper we did not relate to the quality of the exponential family approximation of the data’s distribution.
>
> - Application to deep learning: We thank the referee for pointing the lack of further discussion. We have clarified that in this we mean that current applications of the $\rm{IB}$ in the DL regime can be done using the new $\rm{dualIB}$ formalism, with the latter conserving natural parametrization and symmetries of the original problem.
>
> - Deterministic IB: The Deterministic IB aims to better capture the notion of compression whereas our framework focuses on the parametrization of the data and prediction, thus they concern a different task.
>
> - “preserves the low dimensional sufficient statistics of the data":  As stated by the reviewer the $\rm{dualExpIB}$ decoder (and encoder) are now of exponential form and given by sufficient statistics which are given by the one’s of the original distribution, $A_{r}(x)$, as given in equation 16.  Thus it is given that the dimensionality is preserved, with $d$ such variables. The dependance on $\hat{x}$ is obtained by considering the expectations, $\sum_{x} A_{r}(x) p_{\beta}(x\mid\hat{x})$, of the same functions, hence preserves the values mean.
>
> We realize that the $\rm{dualIB}$ framework is currently given from a theoretical point of view, yet we hope that having made the motivation for it along with it’s advantages clearer now, including the addition of an experiment visualizing superiority of it, that the reviewer finds it relevant to the ICLR conference.

---

### Author Response · Authors · 2019-11-15
**A general response clarifying revisions**

We thank the reviewers for their thoughtful reviews. While we were glad to see that generally the conceptual idea behind the new Dual Information Bottleneck ($\rm{dualIB}$) framework was conveyed we feel that that the reviewers pointed out a major flaw in out presentation - lack of properly explaining the motivation behind the dualIB framework and from there on it's superiority over the original information bottleneck ($\rm{IB}$).

We have thus modified the text as follows:
- Revised the abstract
- Added a subsection to the introduction, sec 1.5 “drawbacks of the IB”.
- Extended the discussion in sec. 1.6 “contributions of this work”.
- Performed an additional experiment specifically relating to the suggested advantages (see sec. 3, Figure 4) of our framework.

With that, we would like to bring the gist of the motivation here as well:

The main theoretical and practical drawback of the $\rm{IB}$ is its reliance on the explicit knowledge of the joint distribution between the input variable $X$ and label $Y$, $p(x,y)$, in a completely non-parametric manner. Often times, the data has natural parametrization (for example biological data-set that is known to follow an analytical model) or that given a finite sample of the data assumptions regarding the "features" can be made, finding the sufficient statistics representation.
Such parametrization allows for a dimensionality reduction (instead of tracking the entire joint distribution $p(x,y)$ we may track the sufficient statistics) and for generalization to patterns we haven’t observed in the given sample.
Having that the IB is non-parametric, our motivation was to find an approximation to it that uses this information to (a) simplify the task and (b) obtain a better generalization as it follows the inner structure of the distribution.
Given this aim, by requiring that the decoder will preserve a parametric structure a new rate-distortion problem has emerged, the Dual Information Bottleneck ($\rm{dualIB}$).
As we show in the text the framework is applicable to any problem; obtaining a good approximation to the IB and it follows the same characteristic behaviour.
The dualIB's superiority emerges in the case that the rule distribution, $p(y|x)$, can be approximated by an exponential family.
In the exponential dual information bottleneck scenario ($\rm{dualExpIB}$) we show that the obtained solutions, encoder-decoder, follow the same exponential form with the parameters defined by the original parameters through expectations. This achieves the desired property of preserving the natural form of the distribution thus obtaining a better generalization, as we now emphasize by providing new experimental results.

---

### Decision · Program_Chairs · 2019-12-19

**Decision:**

Reject

**Comment:**

Main content:

Blind review #1 summarizes it well:

This paper introduces a variant of the Information Bottleneck (IB) framework, which consists in permuting the conditional probabilities of y given x and y given \hat{x} in a Kullback-Liebler divergence involved in the IB optimization criterion.

Interestingly, this change only results in changing an arithmetic mean into a geometric mean in the algorithmic resolution.

Good properties of the exponential families (existence of non-trivial minimal sufficient statistics) are preserved, and an analysis of the new critical points/information plane induced is carried out.

--

Discussion:

The reviews generally agree on the elegant mathematical result, but are critical of the fact that the paper lacks any empirical component whatsoever.

--

Recommendation and justification:

The paper would be good for ICLR if it had any decent empirical component at all; it is a shame that none was presented as this does not seem very difficult.